# Follow the money: Investigating gender disparity in industry payments among senior academics and leaders in plastic surgery

**Ledibabari M. Ngaage**[1], **Chelsea Harris**[2], **Wilmina Landford**[3], **Brooks J. Knighton**[1], **Talia Stewart**[4], **Shealinna Ge**[1], **Ronald P. Silverman**[1,5], **Sheri Slezak**[1], **Yvonne M. Rasko**[1]*

1 Division of Plastic Surgery, Department of Surgery, University of Maryland Medical Center, Baltimore, Maryland, United States of America, 2 Department of Surgery, University of Maryland Medical Center, Baltimore, Maryland, United States of America, 3 Department of Plastic & Reconstructive Surgery, John Hopkins University School of Medicine, Baltimore, Maryland, United States of America, 4 Geisel School of Medicine at Dartmouth, Hanover, New Hampshire, United States of America, 5 Acelity Corporation, San Antonio, TX, United States of America

* yrasko@som.umaryland.edu

**Data Availability Statement:** There are ethical restrictions on sharing a de-identified data set, because data contain potentially identifying and

## Abstract

### Introduction

Differences in academic qualifications are cited as the reason behind the documented gender gap in industry sponsorship to academic plastic surgeons. Gendered imbalances in academic metrics narrow among senior academic plastic surgeons. However, it is unknown whether this gender parity translates to industry payments.

### Methods

We conducted a cross-sectional analysis of industry payments disbursed to plastic surgeons in 2018. Inclusion criteria encompassed (i) faculty with the rank of professor or a departmental leadership position. Exclusion criteria included faculty (i) who belonged to a speciality besides plastic surgery; (ii) whose gender could not be determined; or (iii) whose name could not be located on the Open Payment Database. Faculty and title were identified using departmental listings of ACGME plastic surgery residency programs. We extracted industry payment data through the Open Payment Database. We also collected details on H-index and time in practice. Statistical analysis included odds ratios (OR) and Pearson's correlation coefficient (R).

### Results

We identified 316 senior academic plastic surgeons. The cohort was predominately male (88%) and 91% held a leadership role. Among departmental leaders, women were more likely to be an assistant professor (OR 3.9, $p = 0.0003$) and heads of subdivision (OR 2.1, $p = 0.0382$) than men. Industry payments were distributed equally to male and female senior plastic surgeons except for speakerships where women received smaller amounts compared to their male counterparts (median payments of $3,675 vs $7,134 for women and

sensitive information as determined by the University of Maryland Institutional Review Board. Request for data may be made to the University of Maryland Institutional Review Board at hrpo@umaryland.edu.

**Funding:** The authors received no specific funding for this work.

**Competing interests:** The authors of this paper have read the journal's policy and have the following competing interests: RPS is a paid employee of Acelity Corporation. This does not alter our adherence to PLOS ONE policies on sharing data and materials. There are no patents, products in development or marketed products associated with this research to declare.

men respectively, $p<0.0001$). Career length and H-index were positively associated with dollar value of total industry payments (R = 0.17, $p = 0.0291$, and R = 0.14, $p = 0.0405$, respectively).

## Conclusion

Disparity in industry funding narrows at senior levels in academic plastic surgery. At higher academic levels, industry sponsorship may preferentially fund individuals based on academic productivity and career length. Increased transparency in selection criteria for speakerships is warranted.

## Introduction

The current climate of government austerity and reduced institutional funding [1] has augmented reliance on industry sponsorship as a means for research and innovation. Industry-surgeon collaborations can create opportunities that encourage scholarly impact, recognition, and invited speaker engagements [2–4]. For example, research funding has been shown to positively impact scholarly productivity [5–7], whereas sponsored speakerships reflect a surgeon's expertise and status as a key opinion leader [8]. However, female academic surgeons face an industry pay gap [9–12]. Within academic plastic surgery, female surgeons receive fewer industry payments and a lower monetary value per payment compared to their male counterparts [11]. Furthermore, lack of funding is often cited by female plastic surgeons as a negative impactor on their ability to publish [5]. Lower scholarly impact [2, 13] and underrepresentation among invited speakers in plastic surgery [14] may augment existing gender disparities in academic promotion and leadership. Industry sponsorship has been shown to be influenced by academic productivity and experience [9–11] and the gendered differences in these metrics have been used to explain the observed disparities in industry funding in academic plastic surgery [11].

However, the literature demonstrates that the disparities in academic achievement narrow as academic seniority increases [10, 15], i.e. women in senior leadership positions or higher academic ranks show similar academic qualifications as their male peers. Yet, it is unknown whether gender parity exists in industry payments to senior academic plastic surgeons and departmental leadership. Additionally, no studies have assessed the influence of gender on the nature of industry payment.

This paper aims to 1) characterise senior academics in plastic surgery who received industry sponsorship; and 2) examine the relationship between industry sponsorship andgender, academic rank, leadership position, career length, and academic productivity.

## Methods

### Study population

In November 2018, a list of integrated and independent Plastic Surgery residency programs throughout the United States was accessed from the Accreditation Council for Graduate Medical Education (ACGME) website (https://www.acgme.org/). A list of faculty members and their academic positions was extracted from the official websites of each residency training program. Faculty gender was ascribed based on name, posted photographs, and gendered pronouns on the program website.

## Metrics of professional success

One investigator (BJK) assessed academic rank, and leadership designation (e.g. division head) which was then checked by an independent investigator (LMN). We included faculty with the academic rank of professor or who held a leadership position. Plastic surgeons with departmental leadership roles were included, regardless of their academic position. We categorised the leadership designations into three groups: Chairs and Chiefs, Program Directors, and Heads of Subdivisions. Chairs and chiefs included department chairs and when plastic surgery was a division, division chiefs were also included within this group. Program directors included full and associated directors of plastic surgery residency (integrated and independent) and fellowship programs. Whereas subdivision leadership included those in charge of clinical care programs, education, and/or research for the department. (See S1 Table for a detailed list of the leadership designations within each category).

We utilised the H-index as a proxy measure for academic productivity. The H-index is a bibliometric tool that takes into account the quantity and quality of publications [16]. We identified through the H-index through the Author Search function on the Scopus website (https://www.scopus.com) in December 2018. In cases where multiple H-indices were available for the same physician, the H-index was calculated manually. Career length was defined as time since board certification as identified from the American Board of Plastic Surgery website (https://www.abplasticsurgery.org).

## Industry contributions

The CMS Open Payment database was used to identify industry contributions for each faculty member (https://openpaymentsdata.cms.gov). Identity was confirmed by physician name, specialty and practice location. We recorded the value amounts of all industry payments to academic plastic surgery leaders in 2018. Surgeon engagement with industry was defined as receipt of one or more industry transactions. Details were also collected on type of payment, such as food and beverage, royalties, consulting fees, speaker fees, and payments for educational purposes (See S2 Table for the CMS definitions of each payment type [17]). Inclusion criteria encompassed (i) faculty with the rank of professor or a leadership position. Exclusion criteria included faculty (i) who belonged to a speciality besides plastic surgery; (ii) whose gender could not be determined; or (iii) whose name could not be located on the CMS database.

## Data analysis

Composite data was stored in Microsoft Excel (Microsoft 2016, Redmond, Washington). Data analysis was completed using IBM SPSS Software Version 25.0 (IBM Corp, 2018. IBM SPSS Statistics for Windows, Armonk, NY: IBM Corp). Data were analysed using the Kolmogorov-Smirnov test; H-index and years in practice did not follow a normal distribution so median values and interquartile range (IQR) were reported. We utilised the Mann Whitney-U test to measure differences in non-parametric data when subgroups size was greater than 5 values. Univariate analysis was used to compare gender-based differences in academic rank, H-indices, and time in practice, using Fisher's exact and chi-square tests, as appropriate. The odds ratio was reported with the 95% confidence interval (CI). Linear regression was used to compare differences in industry payment while controlling for academic rank, H-index, and time in practice. Pearson's correlation coefficient (R) is the measure of the linear correlation between two variables. We evaluated multicollinearity, the phenomenon where one predictor variable in a multiple regression model can be linearly predicted from the others, using variance inflation factor (VIF), where VIF = 1 signifies no correlation, VIF 1–5 denotes moderate

correlation, and VIF $>5$ signifies high correlation [18, 19]. Statistical significance was determined to have been achieved for two-tailed value of $p \leq 0.05$.

## Results

Based on the exclusion criteria, 316 academic plastic surgeons were identified. The cohort was predominately male (88%), more academically productive (median H-index: 16 [IQR: 10–23]) and been in practice for 18 years [IQR: 11–25]. Nine percent of the cohort ($n = 28$) were of full professor rank but did not hold an additional departmental leadership role (Table 1). Amongst the 288 surgeons with a leadership position in plastic surgery, women were more likely to be an assistant professor (OR 3.9, 95% CI: 1.87–8.30, $p = 0.0003$) and heads of subdivisions (OR 2.1, 95% CI: 1.04–4.11, $p = 0.0382$). Additionally, women were less academically productive (H-index 11 vs H-index 16, $p = 0.0024$), and had a shorter career length (12 years vs 18 years, $p = 0.0096$).

### Industry payments

The majority of the cohort received a form of industry contribution (82%, $n = 259$). We compared the characteristics of those who received industry contributions to those who did not (Table 2). Plastic surgeons who did not receive any industry contributions had a greater proportion of full professors (82% vs 61%, $p = 0.0021$) and a lower percentage of assistant professors (2% vs 19%, $p = 0.0013$) than those who received industry sponsorship. Additionally, the cohort that did not receive industry contributions had a significantly greater scholarly profile (median H-index 18 vs median H-index 15, $p = 0.0238$) and more experience (22 years vs 16.5 years, $p = 0.0003$) than plastic surgeons who received industry payments.

Similar proportions of male and female senior plastic surgeons engaged with industry (82% vs 82%). This did not significantly differ when the groups were stratified by academic rank (Fig 1A) or leadership position (Fig 1B). Furthermore, the median dollar value of all industry contributions did not differ significantly between male and female senior plastic surgeons ($1074 vs $927, $p = 0.4777$). In contrast, the median dollar value of industry payments differed by academic rank and leadership position, with female associate professors and female subdivision heads receiving the highest total value payments (Fig 2A and 2B). However, no significance was found.

**Table 1. Characteristics of senior academics and departmental leaders in plastic surgery stratified by gender.**

| Characteristic | Male ($n$ = 278) | Female ($n$ = 38) | $p$-value* |
|---|---|---|---|
| Academic rank | | | |
| Assistant Professor | 36 (13%) | 14 (37%) | **0.0006** |
| Associate Professor | 56 (20%) | 5 (13%) | 0.4274 |
| Professor | 186 (67%) | 19 (50%) | 0.0599 |
| Leadership position | | | |
| Chairs and Chiefs | 74 (27%) | 8 (21%) | 0.5967 |
| Program Directors | 60 (22%) | 13 (34%) | 0.1237 |
| Subdivision Heads | 111 (40%) | 22 (58%) | 0.0516 |
| Median H-index | 16 [IQR: 10–23] | 11 [IQR: 5–18] | **0.0026** |
| Median career length / years | 18 [IQR: 11–26] | 12 [IQR: 7–22] | **0.0071** |

**Bold text** denotes statistical significance. * Chi-square test is used to compare proportions and the Mann Whitney U test is used to compared median values.

**Table 2. Characteristics of senior academics and departmental leaders in plastic surgery stratified by receipt of industry payment.**

| Characteristic | Received industry payment (*n* = 259) | Received no industry payment (*n* = 57) | *p*-value* |
|---|---|---|---|
| Academic rank | | | |
| Assistant Professor | 49 (19%) | 1 (2%) | **0.0013** |
| Associate Professor | 52 (20%) | 9 (16%) | 0.4583 |
| Professor | 158 (61%) | 47 (82%) | **0.0021** |
| Leadership position | | | |
| Chairs and Chiefs | 66 (25%) | 16 (28%) | 0.6892 |
| Program Directors | 65 (25%) | 8 (14%) | 0.0727 |
| Subdivision Heads | 115 (44%) | 18 (32%) | 0.0759 |
| Median H-index | 15 [IQR: 9–22.3] | 18 [IQR: 13–25.5] | **0.0238** |
| Median career length / years | 16.5 [IQR: 10–25] | 22 [IQR: 17–27] | **0.0003** |

**Bold text** denotes statistical significance. * Chi-square test is used to compare proportions and the Mann Whitney U test is used to compared median values.

Industry payments were further analysed in respect to nature of payment as defined by CMS (See S2 Table for the CMS definitions of each payment type [18]). Associated research payments were the highest value of all industry contribution types. Of the general industry payments, food was the most common type (92%, *n* = 238) yet only accounted for 3% of the value of all industry payments received. Whereas, royalties had the highest median dollar value of industry payments ($19,823, IQR: $2,975–405,998) and constituted almost a quarter of total dollars received (23%). However, these payments went to eight (3%) senior academic plastic surgeons.

There was no gendered difference in surgeon engagement with industry for different types of payments (Table 3). Similarly, the monetary value of industry contributions was distributed equitably between men and women for all payment types except for speaker fees (Table 3). When all funded speaker events were analysed, women received lower median dollar amounts than their male counterparts ($3,675 vs $7,134, *p*<0.0001). Subgroup analysis of speaker event types lacked sufficient power for meaningful statistical analysis (See S3 Table for the number of recipients and median dollar value of industry payments to senior academic plastic surgeons and departmental leaders stratified by type of payment and gender).

### Regression analysis

Univariate analysis demonstrated that total dollar value of industry contributions was positively associated with academic rank (R = 0.29, *p*<0.0001), possession of a leadership position (R = 0.26, *p*<0.0001), career length (R = 0.19, *p* = 0.0020), and H-index (R = 0.29, *p*<0.0001). Gender was not related to monetary value of industry payments (R = 0.08, *p* = 0.1869).

After adjusting for variables (gender, academic rank, possession of a departmental leadership position, career length, and H-index) through multiple linear regression, career length and H-index remained positively associated with increasing dollar value of total industry payments (R = 0.17, *p* = 0.0291, and R = 0.14, *p* = 0.0405, respectively). Multicollinearity demonstrated low correlation between variables (VIF = 1.2).

### Discussion

Modern plastic surgery is in the midst of a gender revolution. There is an increased female presence at all academic ranks [20, 21], greater female authorship [22, 23], and narrowing of income disparities [24]. Gender parity in industry payments among senior academics and departmental leaders in plastic surgery adds to this growing wave of change. Encouragingly,

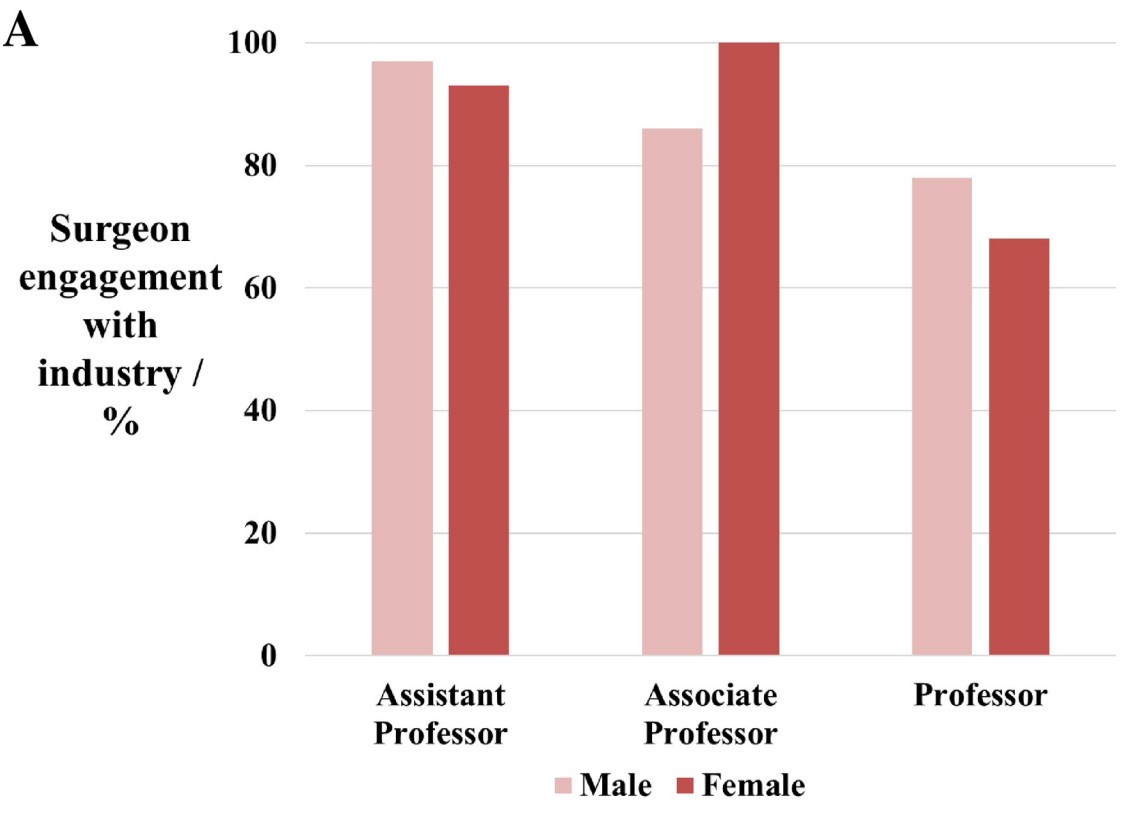

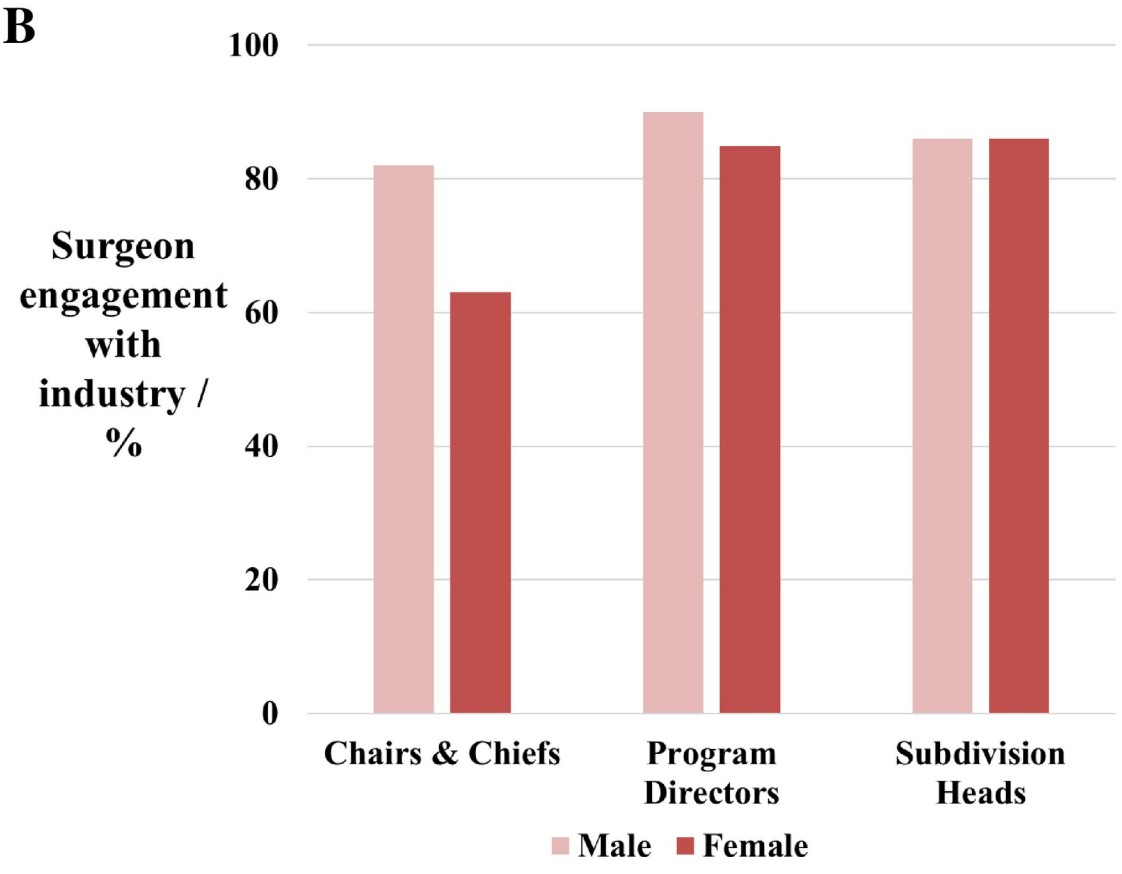

**Fig 1. Male and female senior academics and departmental leaders in plastic surgery stratified by academic rank.** (**A**) Percentage of academic plastic surgeons in receipt of an industry contribution. (**B**) Percentage of academic plastic surgeons in receipt of an industry contribution.

our data suggest that at higher academic levels industry sponsorship is not driven by gender but instead relies on more academic productivity and seniority. However, disparities still exist. In our cohort, men were more likely to be professors and receive larger value sums for speakerships than their female peers.

More encouraging is the evidence that industry sponsorship in academic plastic surgery appears to be gender-neutral. Industry preferentially funds senior surgeons as indicated by career length and academic productivity. This data reflects the established ideology of a threshold effect in which individuals must reach an academic benchmark to qualify for industry funding [9–11]. Given the increased competition for funding, an understanding of the selection criteria employed by companies is needed. Young surgeons seeking to establish financial relationships with industry would benefit from augmenting their research profile. Research expertise is reflected by length of research career and scholarly productivity. Indeed, a recent 2019 study in neurosurgery [10] found that sponsorship was more associated with scholarly productivity when surgeons lacked sufficient career experience. This is supported by the lack of co-linearity between these variables.

Our findings, although promising, show a contrast to the literature which reports a gender gap in industry payments to academic plastic surgeons [11]. We believe this is due to the differences in the group cohorts. Earlier reports assessed gender disparity among all academic plastic surgeons [11]. However, our population consists of only senior plastic surgeons as defined by departmental leadership and rank of full professor. Plastic surgeons in senior academic ranks i.e. full professorship, or with leadership titles, e.g. department chair or division chief, are attractive funding prospects due to their reputation and status. Moreover, they are likely to have met the academic and experience benchmarks required for sponsorship. Given that industry support is associated with markers of academic success [10–12] and gender imbalances in these markers resolve among senior faculty [10, 15], it is rational that that the gender disparity in industry payments narrows accordingly.

Notably, surgeon engagement with industry, regardless of nature of payment, did not differ significantly between male and female academic plastic surgeons. This finding is contrary to the literature which suggests that female physicians have different preferences for industry engagement [25]. However, only men received any payments related to royalties. Compared to women, men are likely to monetise their ideas [26] which may explain the increased prevalence of royalty payments (based on sales of products that use a physician's intellectual property) amongst male academic plastic surgeons. Women may be more likely to focus on teaching than focusing solely on research endeavours [27, 28] which may also be supported by their increased acquisition of subdivision leadership titles.

When stratified by rank, there were no differences between men and women academic plastic surgeons in leadership roles; thus, the differences in average payments are likely owing to the relative seniority of men. Contrary to the literature [6], the median industry contribution across academic rank in our study demonstrated that associate professors garnered a greater amount of support from companies, followed by full professors, and assistant professors. Full professors have already established themselves within their field; they are better published, have longer experience and are considered thought leaders. Therefore, they are more likely to receive governmental funding e.g. the National Institutes of Health [6, 7]. This is supporting by our findings which demonstrated that the cohort who did not receive any industry

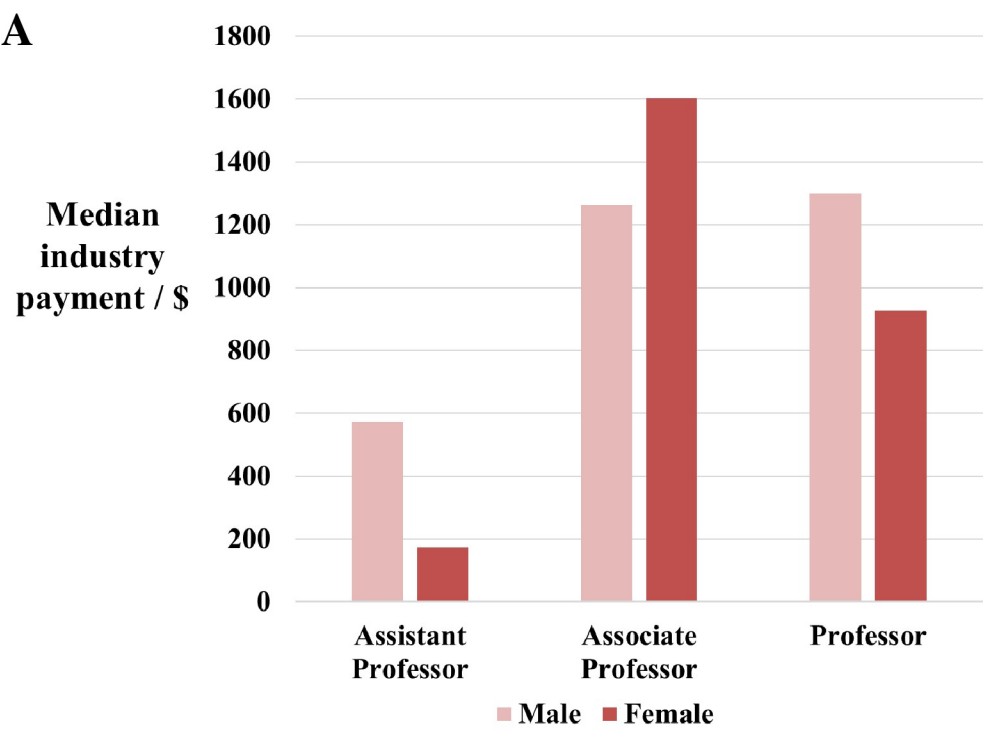

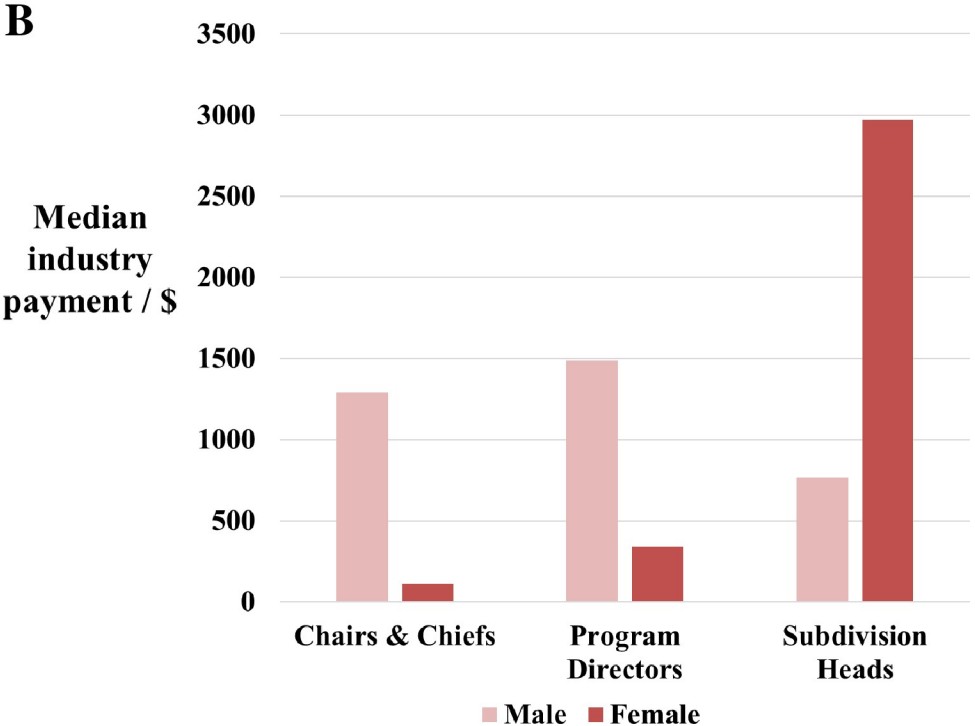

**Fig 2. Male and female academic plastic surgeons stratified by leadership position.** (**A**) Median dollar value of industry contributions to academic plastic surgeons. (**B**) Median dollar value of industry contributions to academic plastic surgeons.

**Table 3. Number of recipients and median dollar value of industry payments to senior academic plastic surgeons and departmental leaders stratified by type of payment and gender.**

| Nature of industry payment | Men (*n* = 228) | Women (*n* = 31) | *p*-value* |
|---|:---:|:---:|:---:|
| Food and Beverage | | | |
| Number of recipients | 211 (92%) | 27 (87%) | 0.5485 |
| Median dollar value | $340 [IQR: 135–748] | $339 [IQR: 100–927] | 0.7039 |
| Consulting | | | |
| Number of recipients | 57 (25%) | 5 (16%) | 0.2774 |
| Median dollar value | $7,650 [IQR: 2,700–21,962] | $6,350 [IQR: 1,925–18,973] | 0.3524 |
| Royalties | | | |
| Number of recipients | 8 (3%) | 0 (0%) | 0.6013 |
| Median dollar value | $19,823 [IQR: 2,975–405,998] | - | - |
| Travel and Accommodation | | | |
| Number of recipients | 66 (29%) | 8 (26%) | 0.7184 |
| Median dollar value | $1,258 [IQR: 692–3,473] | $1,854 [IQR: 293–3,148] | 0.9681 |
| All Speakerships | | | |
| Number of recipients | 39 (17%) | 5 (16%) | 0.8875 |
| Median dollar value | $7,134 [IQR: 2,219–12,758] | $3,675 [IQR: 1,779–30,063] | **<0.0001** |
| Gift | | | |
| Number of recipients | 6 (3%) | 1 (3%) | 0.9892 |
| Median dollar value | $2,700 [IQR: 2,539–11,900] | $18,370 | - |
| Honoraria | | | |
| Number of recipients | 8 (3%) | 0 (0%) | 0.6014 |
| Median dollar value | $1,339 [IQR: 638–3,188] | - | - |
| Grant | | | |
| Number of recipients | 1 (0.4%) | 0 (0%) | 0.9989 |
| Median dollar value | $3,300 | - | - |
| Education | | | |
| Number of recipients | 14 (6%) | 4 (13%) | 0.2457 |
| Median dollar value | $323 [IQR: 87–1,235] | $130 [IQR: 22–313] | - |
| Associated Research | | | |
| Number of recipients | 25 (11%) | 3 (10%) | 0.9879 |
| Median dollar value | $20,586 [IQR: 8,912–41,526] | $60,000 [IQR: 2,040–262,800] | - |

**Bold text** denotes statistical significance.

* Mann Whitney U test is used to compared median values.

payments were more experienced, had a greater scholarly impact, and had a disproportionate representation of full professors. On the other hand, associate professors are still in the process of launching their career. As research funding can positively influence scholarship [29] and therefore plays an important role in promotion, associate professors may be more aggressive in negotiation. Whereas assistant professors, although eager to create a reputation, may be less successful in obtaining sponsorship than associate or full professors.

Our analysis demonstrated that female senior plastic surgeons were more likely to be heads of subdivisions. This is consistent with evidence that women value clinical and educational activities as markers of career success [30]. These roles were previously thought to be less prestigious and less associated with promotion. However, the observed gender equity in total industry contributions suggest a changed perception on the value of these roles. In fact, female heads of subdivisions received larger dollar amounts than other leadership holders.

Subdivision leadership includes those in charge of patient care and research for the department–those at the frontline of innovation. Therefore, we theorise that they are more likely to engage with industry for support. Conversely, chairs and chiefs had the lowest engagement among leadership holders. This may be due to limited time, competing interests, and the concern of the perception of bias that occurs when physicians accept industry contributions [3, 31, 32].

Our findings also suggest that women receive significantly smaller total sums for speaker sponsorship than their male peers. This reinforces previously raised concerns on gender bias in speaker selection [11] and the gross underrepresentation of women among invited speakers in plastic surgery [14]. It is possible that female plastic surgeons are overlooked for industry-funded programs, in addition to the traditional invited speakerships. However, we did not record number of transactions for each payment subtype, so it is possible men are not paid more per transaction but are instead attending more speaking engagements. It is possible that family commitments and pregnancy my limit a female surgeon's ability to travel and, thus, engage in more speakerships. As invited speakership is often a criterion for academic promotion, this may augment gender disparities in promotion and leadership. Given the potential consequences, increased transparency in selection criteria for speakerships is warranted. Future studies may investigate the relationship between sponsorship and speakerships, as well as how gender intersects with these variables.

This study is limited by the small cohort of female plastic surgeons in senior academic rank or leadership roles, the inaccuracies associated with using an online database, and lack of control for other training variables, such as time away for pregnancy and family life. Our cohort was limited to senior academics and departmental leaderships in plastic surgery so there is also potential for selection bias. Additionally, each institution may have different requirements for similarly named positions and promotion criteria may not uniform across institutions. Furthermore, institutions may limit engagement with industry; for example, at our institution, physicians cannot accept more than 10% of their salary unless there are extenuating circumstances. However, it was not possible to control for this confounder within the study.

## Conclusion

There is a changing landscape in academic plastic surgery which has led to a narrowing of gender inequalities among senior levels and departmental leadership. Gender parity in industry payments among senior academics and departmental leaders in plastic surgery adds to this growing wave of change. At higher academic levels, industry sponsorship is not associated with gender and instead preferentially fund individuals based on academic productivity and career length. However, women in academic plastic surgery still face challenges and increased transparency in selection criteria for speakerships is warranted.

## Supporting information

**S1 Table. List of departmental leadership designations within each category.**
(DOCX)

**S2 Table. Definitions of Center for Medicare and Medicaid Services payment types.**
(DOCX)

**S3 Table. Number of recipients and median dollar value of industry-sponsored speakerships to senior academic plastic surgeons and departmental leaders stratified by type of payment and gender.**
(DOCX)

## Author Contributions

**Conceptualization:** Ledibabari M. Ngaage, Wilmina Landford, Ronald P. Silverman, Sheri Slezak, Yvonne M. Rasko.

**Data curation:** Ledibabari M. Ngaage, Brooks J. Knighton, Talia Stewart, Shealinna Ge, Sheri Slezak.

**Formal analysis:** Ledibabari M. Ngaage, Shealinna Ge, Yvonne M. Rasko.

**Investigation:** Brooks J. Knighton, Talia Stewart, Yvonne M. Rasko.

**Methodology:** Ledibabari M. Ngaage, Chelsea Harris, Shealinna Ge, Ronald P. Silverman.

**Resources:** Yvonne M. Rasko.

**Supervision:** Chelsea Harris, Wilmina Landford, Ronald P. Silverman, Sheri Slezak, Yvonne M. Rasko.

**Validation:** Ledibabari M. Ngaage, Chelsea Harris.

**Writing – original draft:** Ledibabari M. Ngaage.

**Writing – review & editing:** Ledibabari M. Ngaage, Chelsea Harris, Wilmina Landford, Brooks J. Knighton, Talia Stewart, Shealinna Ge, Ronald P. Silverman, Sheri Slezak, Yvonne M. Rasko.

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
