## [Decision Letter · Decision Letter 0]

2 Apr 2020

PONE-D-19-30059

Follow the Money: Investigating Gender Disparity in Industry Payments Among Senior Academics and Leaders in Plastic Surgery

PLOS ONE

Dear Dr Rasko,

Thank you for submitting your manuscript to PLOS ONE. After careful consideration, we feel that your well written study has merit but does not fully meet PLOS ONE’s publication criteria as it currently stands. Therefore, we invite you to submit a revised version of the manuscript that addresses the points raised during the review process.

Please, consider especiailly the comments and proposals of reviewer 3. I am convinced that your manuscript will improve enormously. 

We would appreciate receiving your revised manuscript by May 17 2020 11:59PM. To enhance the reproducibility of your results, we recommend that if applicable you deposit your laboratory protocols in protocols.io, where a protocol can be assigned its own identifier (DOI) such that it can be cited independently in the future. For instructions see: http://journals.plos.org/plosone/s/submission-guidelines#loc-laboratory-protocols

We look forward to receiving your revised manuscript.

Kind regards,

Hans-Peter Simmen, M.D., Professor of Surgery

Academic Editor

PLOS ONE

"The authors have declared that no competing interests exist"

We note that one or more of the authors are employed by a commercial company: Acelity Corporation

Reviewers' comments:

Reviewer's Responses to Questions

**Comments to the Author**

1. Is the manuscript technically sound, and do the data support the conclusions?

Reviewer #1: Yes

Reviewer #2: Yes

Reviewer #3: Partly

2. Has the statistical analysis been performed appropriately and rigorously? 

Reviewer #1: Yes

Reviewer #2: Yes

Reviewer #3: Yes

3. Have the authors made all data underlying the findings in their manuscript fully available?

Reviewer #1: Yes

Reviewer #2: Yes

Reviewer #3: Yes

4. Is the manuscript presented in an intelligible fashion and written in standard English?

Reviewer #1: Yes

Reviewer #2: Yes

Reviewer #3: Yes

5. Review Comments to the Author

Reviewer #1: Thank you for your submission of this article. Your study is a database review and comes to reasonable conclusions using standardized measures.

I would like you to point out in either your discussion or conclusions that more and more academic centers in the United States have prohibitions on certain types of industry sponsorships by academic faculty which results in disparities which may be confounding your results. For instance speaker honoraria for non-CME events are often prohibited and this results in certain institutions, most specifically those with the most significant federal research funding, and likely the most gender equality, having the least input into these results.

Reviewer #2: The authors aim to characterise the relationship between industry sponsorship and senior leaders in academic plastic surgery andl evaluate the role of gender on in the nature and monetary value of industry payments. They conclude that ndustry payments were distributed equally to male and female senior plastic surgeons except for speakerships. Career lenght and H-index were positively associated with total industry payments.

The study is clearly written and well performed. I have the following questions

1. How do the authors explain that only a small number of industry grants were included ?

2. Please specify "associated research"

Reviewer #3: General comments:

The study applied a quite strict inclusion and exclusion criteria therefore results in a very limited sample size, especially including only 38 females. It’s unclear how representative is this study sample (as % of academic faculties). The merits of inclusion and exclusion criteria is also questionable. Faculty with the rank of associate professor maybe included, if not those with the rank of assistant professor. Those who processed no payment data for 2018 may also be included as it enables the examination of research question: what are the characteristics of academics who received industry payments.

Measurement of leadership position focuses only on departmental positions. This should be expanded to leadership position in professional societies as well as recipients of major rewards in plastic surgery.

Abstracts

1) Add description of inclusion and exclusion criteria

2) In Results paragraph line 39, change to “for speakerships where women received smaller amounts compared to their male counterparts (median payments of $3,675 vs. $7,134 for women and men respectively, P<0.0001)”.

3) In Results paragraph line 40-42, “Multiple linear regression … (R=0.17, … and R=0.14, …)” - Explain what is the R value.

Introduction

1) Lines 69-71, consider changing to “This paper aims to 1) characterize academics who received industry sponsorship; 2) examine relationship between industry sponsorship and gender, academic rank, leadership position, career length, and academic productivity”.

Method

1) Line 86, consider to include faculty with the academic rank of professor or associate professor.

2) Line 110, consider faculty with the academic rank of professor or associate professor or a leadership position.

3) Lines 112-113, consider deleting “(iv) who possessed no payment data for 2018” from the exclusion criteria.

Data analysis

1) Lines 124-125, explain the meaning of R statistic.

Results

1) Line 133, 316 differs from the sum of 279 and 38 in table 1.

2) Line 140, p=0.0024; Line 141, p=0.0096. Both p-values differ from the p-values in table 1.

3) Lines 147 and 150, re-organize figures: put 1a) and 2a), 1b) and 2b) together.

4) Lines 149-150, “This was also true when the groups were stratified by academic rank (Figure 1b) or leadership position (Figure 2b)”, this interpretation seems to be wrong. Differences between men and women appears to be large, especially in figure 2b) stratified by leadership position.

5) Lines 173, “After adjusting for confounding variables”, explain the list of confounding variables.

Tables

Table 1:

1) Report one p-value from the chi-square test examining the relationship between academic rank and gender, or leadership position and gender.

2) Report statistical tests used to obtain p-values for median H-index and median career length in footnote.

Table 2:

1) Report p-values comparing men and women on median dollar value for education and associated research.

6. PLOS authors have the option to publish the peer review history of their article (what does this mean?). If published, this will include your full peer review and any attached files.

Reviewer #1: Yes: Zubin J. Panthaki, MD, CM, FACS

Reviewer #2: No

Reviewer #3: No

---

## [Author Response · Author response to Decision Letter 0]

23 May 2020

Dear Editor and Reviewers, 

Many thanks for your kind comments and feedback. We have edited the manuscript according to reviewer feedback and included an additional analysis of the cohort who did not receive any industry contributions. We have provided a detailed feedback to each query below. 

Many thanks, 

Yvonne Rasko

General comments:

The study applied a quite strict inclusion and exclusion criteria therefore results in a very limited sample size, especially including only 38 females. It’s unclear how representative is this study sample (as % of academic faculties). The merits of inclusion and exclusion criteria is also questionable. Faculty with the rank of associate professor maybe included, if not those with the rank of assistant professor. Those who processed no payment data for 2018 may also be included as it enables the examination of research question: what are the characteristics of academics who received industry payments. 

Measurement of leadership position focuses only on departmental positions. This should be expanded to leadership position in professional societies as well as recipients of major rewards in plastic surgery.

A: The authors would like to thank you for the review of our manuscript and insightful comments. There are 805 academic plastic surgeons in the US, therefore, our cohort represents 40% of the total population of academic plastic surgeons. Although 38 women is a small sample size, they constitute 12% of our study cohort which is similar to the percentage of female plastic surgeons in the US (16%). The small sample size results from the underrepresentation of women in this surgical specialty and emphasises the urgency and need for this work which highlights other gender disparities within the profession. 

We chose to focus our study on those in senior academic positions in plastic surgery. Earlier studies demonstrate gender disparities in industry payments and academic achievement in plastic surgery. Although the gendered differences in scholarly achievement narrow as one’s academic seniority increases, it is unknown whether the same gender parity exists in industry payments. The field of plastic surgery is relatively small with only four major US professional societies. The leaders of these societies are already included within the cohort as they are also departmental leaders.

The database includes information on physicians who receive no payments from industry i.e. their names can still be found within the database. We have included those within our study and have included an additional section in the results to characterise their details. Those whose names cannot be located in the Open Payment Database are excluded because it is unknown whether they received any industry payments. 

Abstracts

1) Add description of inclusion and exclusion criteria

2) In Results paragraph line 39, change to “for speakerships where women received smaller amounts compared to their male counterparts (median payments of $3,675 vs. $7,134 for women and men respectively, P<0.0001)”.

3) In Results paragraph line 40-42, “Multiple linear regression … (R=0.17, … and R=0.14, …)” - Explain what is the R value.

A: Many thanks for your thoughtful feedback. We have included inclusion and exclusion criteria in the methods. The sentence in the results section has been altered as described. The explanation of the correlation coefficient, R, is included in the methods. 

Introduction

1) Lines 69-71, consider changing to “This paper aims to 1) characterize academics who received industry sponsorship; 2) examine relationship between industry sponsorship and gender, academic rank, leadership position, career length, and academic productivity”. 

A: Many thanks for your comment. We have changed the sentence in the introduction.

Method

1) Line 86, consider to include faculty with the academic rank of professor or associate professor.

2) Line 110, consider faculty with the academic rank of professor or associate professor or a leadership position.

3) Lines 112-113, consider deleting “(iv) who possessed no payment data for 2018” from the exclusion criteria.

A: Thank you for your thorough reviews and feedback. We have edited the sentence within the methods and further clarified our inclusion criteria.

Data analysis

1) Lines 124-125, explain the meaning of R statistic.

A: Many thanks for your comment. The R statistic is Pearson’s correlation coefficient which is use to measure the relationship between variables in a linear regression analysis. We have added this explanation into the data analysis section. 

Results

1) Line 133, 316 differs from the sum of 279 and 38 in table 1. 

2) Line 140, p=0.0024; Line 141, p=0.0096. Both p-values differ from the p-values in table 1.

3) Lines 147 and 150, re-organize figures: put 1a) and 2a), 1b) and 2b) together.

4) Lines 149-150, “This was also true when the groups were stratified by academic rank (Figure 1b) or leadership position (Figure 2b)”, this interpretation seems to be wrong. Differences between men and women appears to be large, especially in figure 2b) stratified by leadership position.

5) Lines 173, “After adjusting for confounding variables”, explain the list of confounding variables.

A: Many thanks for your thoughtful feedback on our manuscript. The sums in the table should be 278 and 38. This has been corrected.

The p values in the text refer to the odds ratios, the p-values in the table refer to the chi-square test. We have added a footnote to the table to explain this. 

We have reorganised the figures as requested.

We have rephrased the sentence within the methods so it now reads: “In contrast, the median dollar value of industry payments differed by academic rank and leadership position, with female associate professors and female subdivision heads receiving the highest total value payments (Figure 2a and 2b). However, no significance was found.”

The variables included in the regression were gender, academic rank, possession of a departmental leadership position, career length, and H-index. We have clarified this within the text of the results.

Tables

Table 1: 

1) Report one p-value from the chi-square test examining the relationship between academic rank and gender, or leadership position and gender.

2) Report statistical tests used to obtain p-values for median H-index and median career length in footnote.

Table 2: 

1) Report p-values comparing men and women on median dollar value for education and associated research.

A: Thank you for reviewing our manuscript and providing detailed feedback. 

We have reported the p-values from a chi-square test examining the relationship between academic rank and gender, and leadership position and gender in Table 1. We have also clarified the statistical test used in a footnote for Table 1.

In Table 2 (now Table 3), the Mann Whitney U test is used to compare the median values amounts. The Mann Whitney U test can only be used if there are at least five samples within each group. Education and associated research have fewer than 5 values in one group which precludes statistical analysis. We have included this explanation within the data analysis section. 

---

## [Editor Report · Decision Letter 1]

9 Jun 2020

Follow the Money: Investigating Gender Disparity in Industry Payments Among Senior Academics and Leaders in Plastic Surgery

PONE-D-19-30059R1

Dear Dr. Rasko,

We’re pleased to inform you that your manuscript has been judged scientifically suitable for publication and will be formally accepted for publication once it meets all outstanding technical requirements.

Kind regards,

Hans-Peter Simmen, M.D., Professor of Surgery

Academic Editor

PLOS ONE
---

## [Editor Report · Acceptance letter]

15 Dec 2020

PONE-D-19-30059R1 

Follow the Money: Investigating Gender Disparity in Industry Payments Among Senior Academics and Leaders in Plastic Surgery 

Dear Dr. Rasko:

I'm pleased to inform you that your manuscript has been deemed suitable for publication in PLOS ONE. Congratulations! Your manuscript is now with our production department. 

Kind regards, 

on behalf of

Dr. Hans-Peter Simmen 

Academic Editor

PLOS ONE